# Prevalence and predictors of work-related depression, anxiety, and stress among waiters: A cross-sectional study in upscale restaurants

Farrukh Ishaque Saah[1]*, Hubert Amu[2], Kwaku Kissah-Korsah[3]

1 Department of Epidemiology and Biostatistics, School of Public Health, University of Health and Allied Sciences, Hohoe, Ghana, 2 Department of Population and Behavioural Sciences, School of Public Health, University of Health and Allied Sciences, Hohoe, Ghana, 3 Department of Population and Health, School of Public Health, University of Cape Coast, Cape Coast, Ghana

☯ These authors contributed equally to this work.
* fsaahpnur14@uhas.edu.gh

**Data Availability Statement:** All relevant data are within the manuscript and its Supporting Information files.

## Abstract

### Background

Poor mental health often interrupts people's regular activities making them unable to work effectively resulting in poor performance and high turnover intention. We examined the prevalence and predictors of depression, anxiety and stress among waiters in upscale restaurants.

### Methods

This descriptive cross-sectional study involved 384 waiters in upscale restaurants in the Accra Metropolis. Data were collected using a pre-tested questionnaire which embedded DASS-21 (Cronbach Alpha = 0.815). The analysis included descriptive and inferential statistics using STATA 15. Statistical significance was set at p-value <0.05 at 95% confidence interval.

### Results

The prevalence of depression was 38.3%, while anxiety and stress were 52.3% and 34.4% respectively. Females (AOR = 1.69, 95%CI = 1.02–2.79), waiters who foresee a better remuneration (AOR = 3.09, 95%CI = 1.95–4.87), consume caffeine (AOR = 1.44, 95%CI = 0.90–2.32), and use non-prescription drugs (AOR = 2.22, 95%CI = 1.39–3.55) were more likely to have depression. Females (AOR = 1.86, 95%CI = 1.17–2.96), those who foresee better remuneration (AOR = 2.85, 95%CI = 1.82–4.49), and those who use non-prescription drugs (AOR = 2.13, 95%CI = 1.38–3.28) were more likely to have anxiety. Females (AOR = 1.74, 95%CI = 1.01–2.99), waiters who are positive of career success (AOR = 1.70, 95%CI = 0.99–2.91), who foresee better remuneration (AOR = 2.99, 95%CI = 1.85–4.83), consume caffeine (AOR = 1.54, 95%CI = 0.93–2.54), and who use non-prescription drugs (AOR = 3.16, 95%CI = 1.93–5.17) were more likely to be stressed.

**Funding:** The authors received no specific funding for this work.

**Competing interests:** The authors have declared that no competing interests exist.

## Conclusion

There is a high prevalence of poor mental health among waiters. Urgent intervention by hospitality stakeholders is needed to improve their working conditions and psychosocial health to accelerate progress towards the Sustainable Development Goal of promoting mental health and wellbeing.

## Introduction

Mental health is critical to achieving the Sustainable Development Goal (SDG) Three which seeks to ensure healthy lives and promote wellbeing for everyone at all ages [1]. In order to attain this, reducing premature death from non-communicable diseases via the prevention and treatment and promotion of mental health is key [1]. "Prevention and treatment of substance abuse, including drug abuse and harmful use of alcohol" is also an inherent part of this goal [2]. Yet, with about 10.7% of the global population suffering from such mental health conditions, poor mental health has become one of the leading causes of disability [3]. Major mental health issues of public health importance in Sub-Saharan Africa (SSA) are depression, anxiety, and stress. For instance, in the general population, one out of every ten (9% and 10%, respectively) have depression and anxiety in SSA [4].

The hospitality industry especially, restaurant service continues to be a significant employer of a substantial proportion of the global population largely due to increasing service patronage worldwide. For instance, globally, about 2.5 billion people patronise restaurant services daily as a result of its convenience and affordability [5]. More people in developing countries continue to eat out daily [6], with an estimated 40% of the day-to-day diet of urban consumers being street foods including restaurants [7]. Similarly, almost all households in Ghana purchase some prepared food away from home while a number of families are almost entirely reliant on street foods such as restaurants [8]. However, imperative to restaurant branding and customer retention as well as attaining competitive advantage in the restaurant industry, providing quality products and services to customers and excellent customer service are key [9, 10]. Thus, waiters are critical to the success of restaurants because healthy employees contribute to good performance [11, 12].

Waiting job in upscale restaurant has impact on the mental health and wellbeing of waiters which subsequently affects their work performance. It predisposes them to health risks including mental health problems. Trivella and colleagues argue that restaurant work can be very stressful and hectic which pose health risks for restaurant staff [13]. Also, the long and anti-social working hours involved in waiting result in 'emotional labour' [14] posing a risk for work-related mental health issues like depression, anxiety and stress [15, 16].

Depression, anxiety and stress usually interrupt people's regular activities resulting in their inability to work effectively and take care of their families [17]. For instance, Dunnagan et al. notes that distressed workers find it difficult to fully utilise their creativity potential [18], such workers are more likely to display poor effectiveness at work [15]. Depression for instance, has been found to hinder productivity and conduce disability, absenteeism, and likelihood of untimely early retirement [19] while distressed workers suffer from a high turnover [20], creating significant economic costs to the business [21]. Agreeably, in the service industry, restaurants continue to have the highest employee turnover [22].

The important role of frontline workers such as waiters in achieving success and maintaining competitive advantage in upscale restaurants cannot be overemphasized. Nevertheless,

little research attention has been given to their health and wellbeing. Most studies in hospitality in Ghana have focused on customer satisfaction and employee turnover [23, 24]. Again, there is a paucity of studies that justify the prevalence of poor mental health (that is, depression, anxiety, and stress) among waiters though many studies acknowledge that the work of waiters negatively impacts on their mental health and general wellbeing. In this study, thus, we examined the prevalence and predictors of work-related mental health conditions among waiters in upscale restaurants. We demonstrated the comorbidity of depression, anxiety and stress, an aspect lacking in literature. The results of this study provide information for evidence-based intervention to tackle the issue of poor mental health among service workers and the general population. This also helps support efforts to achieving the SDG 3.5 target of promoting mental health and wellbeing. Thus, our study assessed the prevalence of depression, anxiety, and stress and their associated factors among waiters in upscale restaurants in Ghana.

## Methods and materials

The study report followed the STROBE guidelines.

### Setting

The study took place in six upscale restaurants in the Accra Metropolis in Ghana. The Metropolis is one of the 16 administrative districts of the Greater Accra region with its capital, Accra, also Ghana's capital. The district is bounded to the North by the Ga West Municipality, to the West by the Ga South Municipality, and to the East by the La Dadekotopon Municipality. On the South, it shares boundary with the Gulf of Guinea. The district has a total population of 1,665,086 (42% of the region's total population) with 48.1% being males while 47.0% are migrants [25]. In addition, 91.2% are Ghanaians by birth while 4.0% are non-Ghanaians. The Metropolis has 89% literacy rate and 52% with the ability to read and write English in addition to some Ghanaian languages [25].

### Study design

This was a descriptive cross-sectional study grounded by the positivist philosophy. The positivist philosophy allowed for the study to make quantifiable observations leading to statistical analyses [26]. The design enabled the study to gather data on a particular phenomenon from a specific population at a point in time using a questionnaire [27].

### Study population and sampling

The study population included waiters working in upscale restaurants in the Accra Metropolitan area. Only waiters who had worked at least three months in an upscale restaurant in the Metropolis were included. Those who were on leave or seriously sick or casual workers were excluded from the study. The sample size was 384 calculated using Cochran's [28] formula:

$\text{n} = \frac{Z^2 p(1-p)}{e^2}$ where n is the sample size, z is value of normal distribution (standard value of 1.96), e is the margin of error (0.05) and p is the assumed prevalence of psychosis (set at 50%).

Waiters from a third of the 18 upscale restaurants (6 hotel restaurants (HR) and 12 standalone restaurants (SR)) in the Metropolis were included. A multistage sampling technique was used. First six restaurants comprising 2 HR and 4 SR were selected by a lottery method. The names of the 6 HR and 12 SR were written on pieces of paper, folded and placed in two separate boxes. The boxes were shaken rigorously and a piece of paper picked at random from each box. A proportionate stratified sampling technique was employed such that 2/3 of the sample were from SR (256) while remaining 128 were from HR. However, the respondents were

selected using a simple random sampling approach that adopted the waiters register at the facilities as sampling frame.

## Procedures

A pre-tested questionnaire (S1 Questionnaire) with a Cronbach Alpha of 0.815 was used for the data collection with support from three trained field staff. The questionnaire comprised of four sections namely; socio-demographics, prospects with upscale restaurants work, substance use, and the 21-item Depression Anxiety Stress Scale (DASS-21, Cronbach Alpha = 0.837–0.863) [29]. Good reliability scores (Cronbach Alpha of 0.94, 0.87, and 0.91) have been reported for the subscales of DASS-21; depression, anxiety, and stress, respectively [30]. In the current study, the reliability scores for depression, anxiety, and stress were 0.890, 0.823, and 0.905 respectively. The questionnaires were self-filled by the respondents, collected and checked for completeness and validity of responses at the end of each data collection session. However, the field staff assisted respondents who found difficulty in reading and understanding the questions. Again, the participants were asked the questions in relation to their work as waiters in order to confirm work-related observations.

## Data analysis

Collected data (S1 Dataset) were entered into EpiData version 4.3 and were exported, cleaned, and analysed with STATA version 15. The analysis included descriptive statistics such as means, frequencies and percentages and inferential statistics such as logistic regression analysis. P-values $<0.05$ were considered statistically significant at 95% confidence interval.

The analysis of DASS-21 component was based on its manual guidelines, and scores from each domain (depression, anxiety and stress) summed up and multiplied by two to make up the original 42-items scale [29]. As such, scores of depression, anxiety, and stress domains greater than 13, 9, and 18 were respectively considered depression, anxiety, and stress cases [31, 32].

The inferential analysis followed first with a binary logistic regression model (Crude Odds Ratio [COR]) to understand the associations existing between the explanatory and outcome variables independently. Consequently, variables that were significant ($p<0.05$) in the first model were used in the second logistic regression model (Adjusted Odds Ratio [AOR]) to explain the associations existing between the explanatory and outcome variables simultaneously.

## Ethical issues

Ethical approval for this study was given by the Ghana Health Service's Research Ethics Committee with approval code of GHS-ERC: 63/05/17. We also obtained permission from the managements of the restaurants before data collection. Prior to including respondents in the study, they were provided with written informed consent following assurance of the highest level of confidentiality and anonymity in information disclosed. This was achieved by removing all personal identification information from the data and the data kept under safe protection of the researchers without access to a third party. However, respondents who tested positive for depression, anxiety or stress using the DASS-21 were informed to seek professional advice and care.

## Results

### Socio-demographic characteristics of respondents

The socio-demographic characteristics of the respondents in the study is presented in Table 1. Most of the 384 waiters (69.5%) were females. Majority (58.3%) were aged between 20–24 years, 70.3% were single and 83.1% were Christians. Also, most of the respondents (72.4%)

**Table 1. Socio-demographic characteristics of respondents.**

| Socio-demographic variable | Frequency | Percentage (%) |
|---|---|---|
| **Sex** | | |
| Male | 117 | 30.5 |
| Female | 267 | 69.5 |
| **Age (completed years)** | **Mean = 23.03±3.8** | |
| <20 | 55 | 14.3 |
| 20–24 | 224 | 58.3 |
| 25–29 | 87 | 22.7 |
| 30+ | 18 | 4.7 |
| **Marital status** | | |
| Single | 270 | 70.3 |
| Married | 114 | 29.7 |
| **Religion** | | |
| Christian | 319 | 83.1 |
| Muslim | 65 | 16.9 |
| **Highest Educational level** | | |
| JHS/JSS | 27 | 7.0 |
| SHS/SSS/A'level/O'level | 278 | 72.4 |
| Tertiary | 79 | 20.6 |
| **Ethnicity** | | |
| Akan | 185 | 48.2 |
| Mole-Dagbani | 36 | 9.4 |
| Ewe | 74 | 19.3 |
| Ga/Dangme | 62 | 16.1 |
| Other | 27 | 7.0 |
| **Years working as a waiter** | | |
| <1 year | 119 | 31.0 |
| 1–5 years | 220 | 57.3 |
| 6–10 years | 37 | 9.6 |
| >10 years | 8 | 2.1 |
| **Years working in current facility** | | |
| <1 year | 163 | 42.4 |
| 1–5 years | 209 | 54.4 |
| 6–10 years | 10 | 2.6 |
| >10 years | 2 | 0.5 |
| **Role at restaurant** | | |
| Headwaiter | 115 | 29.9 |
| Stationed waiter | 242 | 63.1 |
| Supervisor | 27 | 7.0 |

had SHS/A'level/O'level education and 20.3% had Tertiary education. A relative majority (48.2%) were Akan. While 57.3% had worked as waiters for 1–5 years, 54.4% had worked at their present restaurants for 1–5 years. Majority of the respondents (63.1%) were stationed waiters and 29.9% were headwaiters.

## Prospects associated with upscale restaurant work

As shown in Table 2, majority of the respondents did not foresee the likelihood of extended work engagement (56.0%) and better remuneration (63.5%). However, most of them were

**Table 2. Prospects with waiting at upscale restaurant.**

| Variable | Frequency | Percentage (%) |
|---|---|---|
| **Positive about career success** | | |
| No | 118 | 30.7 |
| Yes | 266 | 69.3 |
| **Potential of extended work involvement** | | |
| No | 215 | 56.0 |
| Yes | 169 | 44.0 |
| **Foresee better remuneration** | | |
| No | 244 | 63.5 |
| Yes | 140 | 36.5 |
| **Anticipate getting an advantage for higher roles/position** | | |
| No | 120 | 31.3 |
| Yes | 264 | 68.8 |

positive of their career success (69.3%) and expected gaining advantage for higher roles/position (68.8%) in the restaurant.

## Substance use among waiters in upscale restaurants

Table 3 presents the prevalence of substance use among waiters in upscale restaurants within the last 30 days. It shows that prevalence of non-prescription drug use, caffeine and alcohol consumption were 43.2%, 46.6% and 19.3% respectively whereas marijuana use was 1.6%.

## Prevalence of depression, anxiety and stress among waiters in upscale restaurant

As shown in Fig 1, the prevalence of depression, anxiety and stress among the respondents were 38.3%, 52.3%, and 34.4%, respectively. Overall, 59.6% of the respondents had at least a mental health problem with 24.7% having all the three conditions.

**Table 3. Substance use among waiters in upscale restaurants.**

| Variable | Frequency | Percentage (%) |
|---|---|---|
| **Non-prescription drug use** | | |
| No | 218 | 56.8 |
| Yes | 166 | 43.2 |
| **Caffeine consumption** | | |
| No | 205 | 53.4 |
| Yes | 179 | 46.6 |
| **Alcohol consumption** | | |
| No | 310 | 80.7 |
| Yes | 74 | 19.3 |
| **Cigarette smoking** | | |
| No | 373 | 97.1 |
| Yes | 11 | 2.9 |
| **Marijuana use** | | |
| No | 378 | 98.4 |
| Yes | 6 | 1.6 |

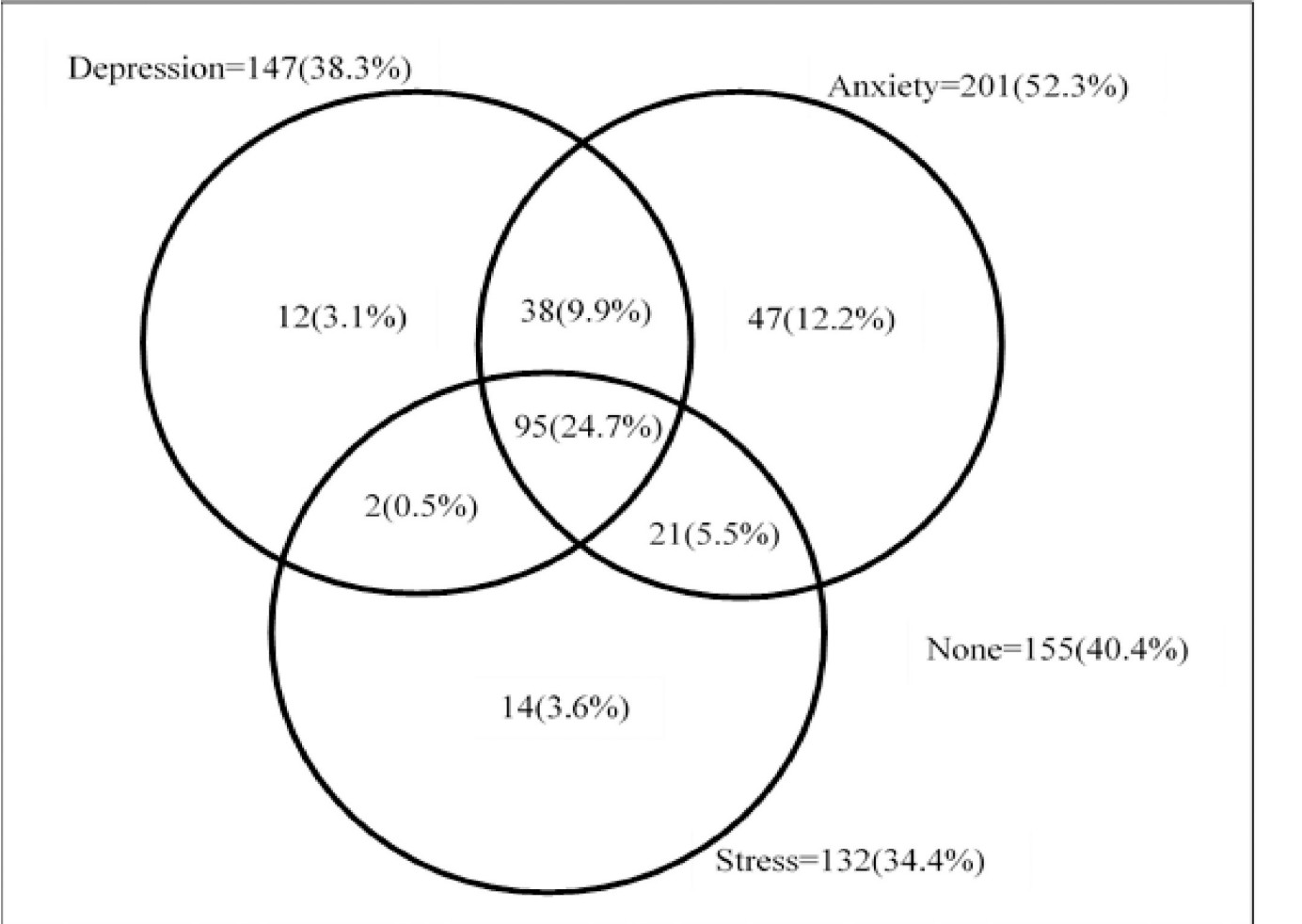

**Fig 1. Prevalence of mental health conditions among waiters.**

### Predictors of depression among waiters in upscale restaurants

Table 4 presents the predictors of depression among waiters in upscale restaurants. Females were 1.69 times (95%CI = 1.02–2.79, p = 0.041) more likely to have depression than males. Compared to waiters who were Akan, those who were Mole-Dagbani (AOR = 1.05, 95%CI = 0.43–2.55, p = 0.918), Ewe (AOR = 2.68, 95%CI = 0.85–8.50, p = 0.093), Ga-Dangme (AOR = 1.50, 95% CI = 0.57–3.98, p = 0.412), and other ethnicity (AOR = 1.02, 95%CI = 0.37–2.78, p = 0.975) were more likely to have depression. While waiters who foresee better remuneration were 3.09 times (95%CI = 1.95–4.87, p<0.001) more likely to be depressed than those who do not, waiters who consume caffeine were 1.44 times (95%CI = 0.90–2.32, p = 0.130) more likely to have depression than those who do not consume caffeine. Waiters who use non-prescription drugs were 2.22 times (95%CI = 1.39–3.55, p = 0.001) more likely to be depressed than those who do not use.

### Predictors of anxiety among waiters in upscale restaurants

Table 5 shows the predictors of anxiety among waiters in upscale restaurants. Females were 1.86 times (95%CI = 1.17–2.96, p = 0.009) more likely to have anxiety than males, whereas

**Table 4. Predictors of depression among waiters.**

| Variable | Depression level | | COR(95%CI)p-value | AOR(95%CI)p-value |
|---|---|---|---|---|
| | Depressed n(%) | Normal n(%) | | |
| **Sex** | | | | |
| Male | 36(30.8) | 81(69.2) | Ref | Ref |
| Female | 111(41.6) | 156(58.4) | 1.60(1.01–2.54)0.046* | 1.69(1.02–2.79)0.041* |
| **Age (in completed years)** | | | | |
| <20 | 35(63.6) | 20(36.4) | | |
| 20–24 | 114(50.9) | 110(49.1) | 1.03(0.35–3.02)0.952 | |
| 25–29 | 44(50.6) | 43(49.4) | 1.31(0.50–3.44)0.586 | |
| 30+ | 8(44.4) | 10(55.6) | 1.52(0.54–4.26)0.425 | |
| **Marital status** | | | | |
| Single | 109(40.4) | 161(59.6) | Ref | |
| Married | 38(33.3) | 76(66.7) | 0.74(0.47–1.17)0.196 | |
| **Religion** | | | | |
| Christian | 127(39.8) | 192(60.2) | Ref | |
| Muslim | 20(30.8) | 45(69.2) | 0.67(0.38–1.19)0.173 | |
| **Highest Educational level** | | | | |
| JHS/JSS | 9(33.3) | 18(66.7) | Ref | |
| SHS/SSS/A'level/O'level | 102(37.4) | 174(62.6) | 1.51(0.61–3.78)0.377 | |
| Tertiary | 34(43.0) | 45(57.0) | 1.26(0.76–2.10)0.365 | |
| **Ethnicity** | | | | |
| Akan | 75(40.5) | 110(59.5) | Ref | Ref |
| Mole-Dagbani | 9(25.0) | 27(75.0) | 1.58(0.70–3.55)0.269 | 1.05(0.43–2.55)0.918 |
| Ewe | 25(33.8) | 49(66.2) | 3.23(1.11–9.39)0.031* | 2.68(0.85–8.50)0.093 |
| Ga-Dangme | 24(38.7) | 38(61.3) | 2.11(0.86–5.17)0.102 | 1.50(0.57–3.98)0.412 |
| Other | 14(51.9) | 13(48.1) | 1.71(0.69–4.24)0.251 | 1.02(0.37–2.78)0.975 |
| **Years working as a waiter** | | | | |
| < 1 year | 43(36.1) | 76(63.9) | Ref | |
| 1–5 years | 85(38.6) | 135(61.4) | 1.77(0.42–7.43)0.437 | |
| 6–10 years | 15(40.5) | 22(59.5) | 1.59(0.39–6.52)0.521 | |
| >10 years | 4(50.0) | 4(50.0) | 1.47(0.32–6.80)0.624 | |
| **Years working in current facility** | | | | |
| < 1 year | 59(36.2) | 104(63.8) | Ref | |
| 1–5 years | 80(38.3) | 129(61.7) | 1.76(0.11–28.70)0.690 | |
| 6–10 years | 7(70.0) | 3(30.0) | 1.61(0.10–26.14)0.737 | |
| >10 years | 1(50.0) | 1(50.0) | 0.43(0.02–9.36)0.590 | |
| **Role at restaurant** | | | | |
| Headwaiter | 45(39.1) | 70(60.9) | Ref | |
| Station waiter | 94(38.8) | 148(61.2) | 0.66(0.26–1.62)0.360 | |
| Supervisor | 8(29.6) | 19(70.4) | 0.66(0.28–1.58)0.352 | |
| **Positive of career success** | | | | |
| No | 44(37.3) | 74(62.7) | Ref | |
| Yes | 103(38.7) | 163(61.3) | 1.06(0.68–1.66)0.790 | |
| **Potential of extended work** | | | | |
| No | 85(39.5) | 130(60.5) | Ref | |
| Yes | 62(36.7) | 107(63.3) | 0.89(0.59–1.34)0.569 | |
| **Foresee better remuneration** | | | | |
| No | 68(27.9) | 176(72.1) | Ref | Ref |

*(Continued)*

**Table 4.** (Continued)

| Variable | Depression level | | COR(95%CI)p-value | AOR(95%CI)p-value |
|---|---|---|---|---|
| | Depressed n(%) | Normal n(%) | | |
| Yes | 79(56.4) | 61(43.6) | 3.35(2.17–5.18)<0.001*** | 3.09(1.95–4.87)<0.001*** |
| **Anticipate an advantage for higher roles/position** | | | | |
| No | 45(37.5) | 75(62.5) | Ref | |
| Yes | 102(38.6) | 162(61.4) | 1.05(0.67–1.64)0.832 | |
| **Caffeine consumption** | | | | |
| No | 64(31.2) | 141(68.8) | Ref | Ref |
| Yes | 83(46.4) | 96(53.6) | 1.91(1.26–2.89)0.002** | 1.44(0.90–2.32)0.130 |
| **Alcohol consumption** | | | | |
| No | 120(38.7) | 190(61.3) | Ref | |
| Yes | 27(36.5) | 47(63.5) | 0.91(0.54–1.54)0.724 | |
| **Cigarette smoking** | | | | |
| No | 140(37.5) | 233(62.5) | Ref | |
| Yes | 7(63.6) | 4(36.4) | 2.91(0.84–10.13)0.093 | |
| **Marijuana use** | | | | |
| No | 142(37.6) | 236(62.4) | Ref | |
| Yes | 5(83.3) | 1(16.7) | 8.31(0.96–71.85)0.054 | |
| **Non-prescription drug use** | | | | |
| No | 62(28.4) | 156(71.6) | Ref | Ref |
| Yes | 85(51.2) | 81(48.8) | 2.64(1.73–4.03)<0.001*** | 2.22(1.39–3.55)0.001** |

*p<0.05

**p<0.01

***p<0.001 COR-Crude Odds Ratio AOR-Adjusted Odds Ratio

those who foresee better remuneration were 2.85 times (95%CI = 1.82–4.49, p<0.001) more likely to have anxiety than those who do not. Similarly, waiters who use non-prescription drugs were 2.13 times (95%CI = 1.38–3.28, p = 0.001) more likely to have anxiety than those who do not use non-prescription drugs.

## Predictors of stress among waiters in upscale restaurants

Predictors of stress among waiters in upscale restaurant is shown in Table 6. Regarding sex of the waiters, females were 1.74 times (95%CI = 1.01–2.99, p = 0.047) more likely to have stress than males. Waiters who were Muslims were 49% (AOR = 0.51, 95%CI = 0.24–1.02, p = 0.072) less likely to have stress than waiters who were Christians. Waiters who were Mole-Dagbani (AOR = 1.54, 95%CI = 0.61–3.86, p = 0.361), Ewe (AOR = 2.16, 95%CI = 0.67–6.90, p = 0.196), Ga-Dangme (AOR = 1.83, 95%CI = 0.67–5.03, p = 0.242), and other ethnicity (AOR = 1.29, 95%CI = 0.46–3.67, p = 0.629) were more likely to have stress than those who were Akan. Also, waiters who are positive of career success were 1.70 times (95%CI = 0.99–2.91, p = 0.053) more likely to have stress than those who are not. Waiters who foresee better remuneration were 2.99 times (95%CI = 1.85–4.83, p<0.001) more likely to have stress than those who do not while those who consume caffeine were 1.54 times (95%CI = 0.93–2.54, p = 0.093) more likely to have stress than those who do not consume caffeine. Likewise, waiters who use non-prescription drugs were 3.16 times (95%CI = 1.93–5.17, p<0.001) more likely to be stressed than those who do not use non-prescription drugs.

**Table 5. Predictors of anxiety among waiters.**

| Variable | Anxiety level | | COR(95%CI)p-value | AOR(95%CI)p-value |
|---|---|---|---|---|
| | Anxious n(%) | Normal n(%) | | |
| **Sex** | | | | |
| Male | 49(41.9) | 68(58.1) | Ref | Ref |
| Female | 152(56.9) | 115(43.1) | 1.83(1.18–2.85)0.007** | 1.86(1.17–2.96)0.009** |
| **Age (in completed years)** | | | | |
| <20 | 35(63.6) | 20(36.4) | Ref | |
| 20–24 | 114(50.9) | 110(49.1) | 0.46(0.16–1.35)0.155 | |
| 25–29 | 44(50.6) | 43(49.4) | 0.77(0.29–2.03)0.599 | |
| 30+ | 8(44.4) | 10(55.6) | 0.78(0.28–2.17)0.636 | |
| **Marital status** | | | | |
| Single | 141(52.2) | 129 (74.4) | Ref | |
| Married | 60(52.6) | 54(47.4) | 1.02(0.66–1.58)0.942 | |
| **Religion** | | | | |
| Christian | 173(54.2) | 146(45.8) | Ref | |
| Muslim | 28(43.1) | 37(56.9) | 0.64(0.37–1.09)0.102 | |
| **Highest Educational level** | | | | |
| JHS/JSS | 12(44.4) | 15(55.6) | Ref | |
| SHS/SSS/A'level/O'level | 146(52.5) | 132(47.5) | 1.49(0.62–3.60)0.371 | |
| Tertiary | 43(54.4) | 36(45.6) | 1.08(0.65–1.78)0.764 | |
| **Ethnicity** | | | | |
| Akan | 102(55.1) | 83(44.9) | Ref | |
| Mole-Dagbani | 16(44.4) | 20(55.6) | 1.02(0.45–2.29)0.967 | |
| Ewe | 33(44.6) | 41(55.4) | 1.56(0.57–4.27)0.384 | |
| Ga-Dangme | 35(56.5) | 27(43.5) | 1.55(0.64–3.77)0.331 | |
| Other | 15(55.6) | 12(44.4) | 0.96(0.39–2.40)0.938 | |
| **Years working as a waiter** | | | | |
| < 1 year | 57(47.9) | 62(52.1) | Ref | |
| 1–5 years | 122(55.5) | 98(44.5) | 1.81(0.41–7.93)0.429 | |
| 6–10 years | 17(45.9) | 20(54.1) | 1.34(0.31–5.74)0.694 | |
| >10 years | 5(62.5) | 3(37.5) | 1.96(0.41–9.43)0.401 | |
| **Years working in current facility** | | | | |
| < 1 year | 79(48.5) | 84(51.5) | Ref | |
| 1–5 years | 114(54.5) | 95(45.5) | 1.06(0.07–17.29)0.966 | |
| 6–10 years | 7(70.0) | 3(30.0) | 0.83(0.05–13.50)0.898 | |
| >10 years | 1(50.0) | 1(50.0) | 0.43(0.02–9.36)0.590 | |
| **Role at restaurant** | | | | |
| Headwaiter | 64(55.7) | 51(44.3) | Ref | |
| Station waiter | 124(51.2) | 118(48.8) | 0.74(0.32–1.71)0.482 | |
| Supervisor | 13(48.1) | 14(51.9) | 0.88(0.40–1.96)0.761 | |
| **Positive of career success** | | | | |
| No | 63(53.4) | 55(46.6) | Ref | |
| Yes | 138(51.9) | 128(48.1) | 0.94(0.61–1.45)0.785 | |
| **Potential of extended work** | | | | |
| No | 112(52.1) | 103(47.9) | Ref | |
| Yes | 89(52.7) | 80(47.3) | 1.02(0.68–1.53)0.912 | |
| **Foresee better remuneration** | | | | |
| No | 104(42.6) | 140(57.4) | Ref | Ref |

*(Continued)*

**Table 5.** (Continued)

| Variable | Anxiety level | | COR(95%CI)p-value | AOR(95%CI)p-value |
|---|---|---|---|---|
| | Anxious n(%) | Normal n(%) | | |
| Yes | 97(69.3) | 43(30.7) | 3.04(1.96–4.71)<0.001*** | 2.85(1.82–4.49)<0.001*** |
| **Anticipate an advantage for higher roles/position in current facility** | | | | |
| No | 63(52.5) | 57(47.5) | Ref | |
| Yes | 138(52.3) | 126(47.7) | 0.99(0.64–1.53)0.967 | |
| **Caffeine consumption** | | | | |
| No | 98(47.8) | 107(52.2) | Ref | |
| Yes | 103(57.5) | 76(42.5) | 1.48(0.99–2.22)0.057 | |
| **Alcohol consumption** | | | | |
| No | 166(53.5) | 144(46.5) | Ref | |
| Yes | 35(47.3) | 39(52.7) | 0.78(0.45–1.29)0.334 | |
| **Cigarette smoking** | | | | |
| No | 194(52.0) | 179(48.0) | Ref | |
| Yes | 7(63.6) | 4(36.4) | 1.62(0.47–5.61)0.451 | |
| **Marijuana use** | | | | |
| No | 195(51.6) | 183(48.4) | Ref | |
| Yes | 6(100.0) | 0(0.0) | - | |
| **Non-prescription drug use** | | | | |
| No | 94(43.1) | 124(56.9) | Ref | Ref |
| Yes | 107(64.5) | 59(35.5) | 2.39(1.58–3.63)<0.001*** | 2.13(1.38–3.28)0.001** |

*p<0.05

**p<0.01

***p<0.001 COR-Crude Odds Ratio AOR-Adjusted Odds Ratio

## Discussion

Our study investigated the prevalence and predictors of depression, anxiety, and stress among waiters in upscale restaurants. Most of the waiters did not have positive prospects associated with working in upscale restaurants. We found that majority of the waiters had mental health problems and significant proportion of them consumed caffeine and alcohol as well as used non-prescription drugs. While waiters' sex, foreseeing better remuneration and non-prescription drug use were the common predictors of all the three, marijuana use was a common predictor for depression and anxiety and waiters' caffeine consumption commonly predicted depression and stress.

It was observed that majority of the respondents did not foresee the likelihood of extended work engagement and better remuneration while most of them were positive about their career success and anticipated getting advantage for higher roles/positions in the present restaurant. These findings show that even though Ng and Burke [33] posit that opportunities for progression and good training, together with a good initial salary are the most desirable job and organizational attributes important for waiters, this study found that most waiters are not given such expectations at their current workplace.

Our study also found that prevalence of depression, anxiety and stress were 38.3%, 52.3%, and 34.4%, respectively. Overall, 24.7% of waiters had all the three mental health conditions which is significantly high. The finding that 38.3% of the waiters in our study had depression is higher than that of Shani and Pizam [34] which found that only 8.7% of the hotel employees had work-related depression and Sipsma and colleagues who reported 18.7% of their

**Table 6. Predictors of stress among waiters.**

| Variable | Stress level | | COR(95%CI)p-value | AOR(95%CI)p-value |
|---|---|---|---|---|
| | Stressed n(%) | Normal n(%) | | |
| **Sex** | | | | |
| Male | 30(25.6) | 87(74.4) | Ref | Ref |
| Female | 102(38.2) | 165(61.8) | 1.79(1.11–2.91)0.018* | 1.74(1.01–2.99)0.047* |
| **Age (in completed years)** | | | | |
| <20 | 23(41.8) | 32(58.2) | Ref | |
| 20–24 | 76(33.9) | 148(66.1) | 0.40(0.12–1.37)0.143 | |
| 25–29 | 29(33.3) | 58(66.7) | 0.56(0.18–1.75)0.316 | |
| 30+ | 4(22.2) | 14(77.8) | 0.57(0.17–1.89)0.360 | |
| **Marital status** | | | | |
| Single | 97(35.9) | 173(64.1) | Ref | |
| Married | 35(30.7) | 79(69.3) | 0.79(0.49–1.26)0.325 | |
| **Religion** | | | | |
| Christian | 119(37.3) | 200(62.7) | Ref | Ref |
| Muslim | 13(20.0) | 52(80.0) | 0.42(0.22–0.80)0.009** | 0.51(0.24–1.02)0.072 |
| **Highest Educational level** | | | | |
| JHS/JSS | 8(29.6) | 19(70.4) | Ref | |
| SHS/SSS/A'level/O'level | 94(33.8) | 184(66.2) | 1.45(0.57–3.73)0.436 | |
| Tertiary | 30(38.0) | 49(62.0) | 1.20(0.71–2.01)0.493 | |
| **Ethnicity** | | | | |
| Akan | 61(33.0) | 124(67.0) | Ref | Ref |
| Mole-Dagbani | 11(30.6) | 25(69.4) | 2.54(1.12–5.76)0.026* | 1.54(0.61–3.86)0.361 |
| Ewe | 23(31.1) | 51(68.9) | 2.84(1.01–8.03)0.049* | 2.16(0.67–6.90)0.196 |
| Ga-Dagnme | 22(35.5) | 40(64.5) | 2.77(1.12–6.85)0.027* | 1.83(0.67–5.03)0.242 |
| Other | 15(55.6) | 12(44.4) | 2.27(0.91–5.70)0.080 | 1.29(0.46–3.67)0.629 |
| **Years working as a waiter** | | | | |
| < 1 year | 42(35.3) | 77(64.7) | Ref | |
| 1–5 years | 78(35.5) | 142(64.5) | 1.10(0.25–4.83)0.900 | |
| 6–10 years | 9(24.3) | 28(75.7) | 1.09(0.25–4.69)0.906 | |
| >10 years | 3(37.5) | 5(62.5) | 1.87(0.37–9.40)0.449 | |
| **Years working in current facility** | | | | |
| < 1 year | 55(33.7) | 108(66.3) | Ref | |
| 1–5 years | 72(34.4) | 137(65.6) | - | |
| 6–10 years | 5(50.0) | 5(50.0) | - | |
| >10 years | 0(0.0) | 2(100.0) | - | |
| **Role at restaurant** | | | | |
| Headwaiter | 42(36.5) | 73(63.5) | Ref | |
| Station waiter | 81(33.5) | 161(66.5) | 0.87(0.36–2.11)0.756 | |
| Supervisor | 9(33.3) | 18(66.7) | 0.99(0.43–2.31)0.989 | |
| **Positive of career success** | | | | |
| No | 31(26.3) | 87(73.7) | Ref | Ref |
| Yes | 101(38.0) | 165(62.0) | 1.72(1.06–2.77)0.027* | 1.70(0.99–2.91)0.053 |
| **Potential of extended work** | | | | |
| No | 75(34.9) | 140(65.1) | Ref | |
| Yes | 57(33.7) | 112(66.3) | 0.95(0.62–1.45)0.813 | |
| **Foresee better remuneration** | | | | |
| No | 60(24.6) | 184(75.4) | Ref | Ref |

*(Continued)*

**Table 6.** (Continued)

| Variable | Stress level | | COR(95%CI)p-value | AOR(95%CI)p-value |
|---|---|---|---|---|
| | Stressed n(%) | Normal n(%) | | |
| Yes | 72(51.4) | 68(48.6) | 3.25(2.09–5.05)<0.001*** | 2.99(1.85–4.83)<0.001*** |
| **Anticipate an advantage for higher roles/position** | | | | |
| No | 36(30.0) | 84(70.0) | Ref | |
| Yes | 96(36.4) | 168(63.6) | 1.33(0.84–2.12)0.224 | |
| **Caffeine consumption** | | | | |
| No | 52(25.4) | 153(74.6) | Ref | Ref |
| Yes | 80(44.7) | 99(55.3) | 2.38(1.55–3.66)<0.001*** | 1.54(0.93–2.54)0.093 |
| **Alcohol consumption** | | | | |
| No | 109(35.2) | 201(64.8) | Ref | |
| Yes | 23(31.1) | 51(68.9) | 0.83(0.48–1.43)0.507 | |
| **Cigarette smoking** | | | | |
| No | 126(33.8) | 247(66.2) | Ref | |
| Yes | 6(54.5) | 5(45.5) | 2.35(0.70–7.86)0.164 | |
| **Marijuana use** | | | | |
| No | 128(33.9) | 250(66.1) | Ref | |
| Yes | 4(66.7) | 2(33.3) | 3.91(0.71–21.61)0.118 | |
| **Non-prescription drug use** | | | | |
| No | 46(21.1) | 172(78.9) | Ref | Ref |
| Yes | 86(51.8) | 80(48.2) | 4.02(2.57–6.28)<0.001*** | 3.16(1.93–5.17)<0.001*** |

*p<0.05

**p<0.01

***p<0.001. COR-Crude Odds Ratio AOR-Adjusted Odds Ratio

respondents having depression [35]. Again, we observed a much higher prevalence of anxiety than in a Malaysian study which found 8.2% having anxiety [36]. However, our stress prevalence does not support a previous study which reported 39% stress prevalence among hotel employees in Malmo [37]. The finding that most of the waiters had at least a psychological issue supports the argument that the hospitality workplace itself may lead to the development of psychological distresses such as depression, anxiety and stress among employees [34]. It is also consistent with the position that service workers are more likely to experience poor psychosocial health [15].

The high prevalence of poor mental health observed among the waiters may also be attributed to waiters' low seniority and status in the restaurant organizational structure [38] together with unfavourable working conditions such as shift and anti-social schedules [15], heavy work strain, low social support at work, and increased psychological demands [19] which are found to accelerate the outburst of depression, anxiety and stress symptoms. Also, our finding could have resulted from excessive workload and working hours, and imbalance between work and social/family life as previously argued [39].

Our study also found that while almost half of the waiters consumed caffeine and non-prescription drug, one in five consumed alcohol. These findings are in congruence to those of previous studies which found that illicit substance use including alcohol and caffeine is prevalent among food service employees [40, 41] with the industry leading in substance use among workers [42]. This is consistent with the argument that caffeine and energy drink consumption is associated with high-demanding jobs as countermeasure for sleep loss and heavy job

demands [43]. Again, our finding that about 20% of the waiters consumed alcohol is lower than that found among hotel employees in a Taiwan surveyed where almost all (90%) consumed some kind of alcoholic drinks with 82·5% using alcohol in the previous month [41]. At the restaurants there is availability of alcohol and caffeine drinks which may have influenced the waiters' consumption of these substances [44]. In addition, substance use such as alcohol, caffeine and non-prescription drugs may result from waiters' effort to alleviate emotional stressors related to their environment [45].

We found that sex of respondent, foreseeing better remuneration and non-prescription drug use were the common predictors of all the three; depression, anxiety, and stress. Nevertheless, our finding that sex influenced waiters' risk of depression, anxiety and stress does not support that of a similar study which found no significant association between hotel employees' sex and depression [34]. This may be due to other social commitments and workload such as caring for children, meeting relation expectations and household chores specially among female waiters.

Also, our finding that job prospects such as foreseeing better remuneration and positivity of career success significantly influencing risk of poor mental health is consistent with the findings of previous studies. Researchers found that service employees experiencing insufficient or general lack of employment security [19, 39], decision-making empowerment [39], balanced effort-reward conditions, and appropriate recognition [46] are at higher risk of mental health problems including symptoms of depression, anxiety and stress.

Our finding that non-prescription drug use was a predictor to depression, anxiety and stress, supports the argument that non-prescription drug may affect the mental health as well as health-related quality of life among the users [47]. This could be attributed to the use of these drugs such as painkillers for symptomatic treatment against body pains, tiredness and general weakness associated with long hours of standing and movement in waiting job. Thus, waiters may have the symptoms for depression, anxiety and stress and resort to using non-prescription drugs for treatment or may develop these symptoms as a result of using these drugs.

Marijuana use was a common significant predictor for both depression and anxiety whereas caffeine consumption and waiters' ethnicity were common significant predictors for both depression and stress. This finding supports the argument that marijuana use is a significant factor to mental health problems especially depression and anxiety [48–50]. However, this finding does not agree with that of other studies which reported that there is no significant association between marijuana use and anxiety even though regular marijuana use has been found to be associated with anxiety symptoms and disorders [51, 52]. Our finding could be attributed to the very few waiters using marijuana in this study possibly to help cope with emotional pain from work-related stressors [43].

Regarding the finding that waiters' ethnicity was a common predictor for depression and stress although not statistically significant, this supports the argument that cultural dynamics and background of individuals influence their mental health [53, 54]. This can be explained to result from the effect of culture on the expectations and assumptions of individuals which affects interpretation of mental health symptoms as well as health seeking behaviours [55], and the differences in cultural practices such as diet and beliefs as espoused in a previous study [56]. The finding also suggests that government policies and interventions to mitigate mental health problems need to be designed and implemented based on ethnic and cultural diversity.

This study, however observed that respondents' religion and positivity about career success were predictors specific to only stress. Religion has been shown to play a significant part in coping with stress and thus, it is supported by our finding that waiters' religion predicted their stress level [57]; however, this was not statistically significant. This is because religion has important impact on stress by either being a cause or cure for stress [58]. Also, regarding

positivity about career success predicting stress among the waiters, this is consistent with the findings of previous studies that one's work significantly impact their mental health and the level of stress among employees is related to their career success and advancement [59, 60]. This could be due to the challenges associated with poor career development and its psychological impact presenting as stressors for such employees recognizing that positive career development is associated with increased salary and other benefits.

## Strengths and limitations

A key strength of this study was our demonstration of the comorbidities of the three mental health conditions, making it possible to exhaustively appreciate the burden of depression, anxiety, and stress as experienced by working individuals in the adult population. Our use of logistic regression also ensured that we robustly established the relationships existing between mental health conditions and the explanatory variables. Also, we used standardized scale, DASS-21, to ascertain depression, anxiety, and stress which is robust and effectively measures the three outcome variables. The study is also one of its kind in the hospitality industry to quantify an issue that is well-acknowledged to be predominant but limitedly investigated. However, reliance on verbal report is the major limitation to the study as this may have resulted in overreporting of socially desirable responses. Even though physical activity is a strong predictor of mental health, it was not included in this study.

## Conclusions

We found a high prevalence of poor mental health and worrying levels of substance use among the waiters. Both substance use and work prospects were found to impact waiters' mental health and wellbeing. These findings have grave implications for the sustainability and growth of the hospitality industry, especially restaurant services as a result of absenteeism, presentism, low performance, and increased turnover likely to occur due to poor health. Also, these findings suggest that Ghana may not be able to achieve the SDG goal 3 requiring that substance abuse and harmful use of alcohol be reduced and ensure promotion of mental health and wellbeing by 2030.

## Recommendations

Our study findings suggest that efforts are needed to ensure that Ghana accelerates progress towards achievement of SDG 3, especially targets 3.4 and 3.5 by the year 2030. Thus, there should be enhanced collaboration between stakeholders in the health and hospitality industries to develop interventions to reduce substance use and harmful alcohol consumption, depression, anxiety, and stress among hospitality workers. The Ministry of Health and the Ghana Health Service in partnership with international organisations such as the World Health Organisation should increase awareness creation and implement interventions targeted at the prevention of mental health problems. Also, the Ministry of Tourism, Trade Union Congress, the Ghana Tourism Authority and the Ghana Tourism Federation should develop policies to improve the working conditions and environment for employees in the hospitality industry and other service sectors.

## Supporting information

**S1 Questionnaire. Questionnaire on depression, anxiety, and stress and its predictors.**
(PDF)

**S1 Dataset. Dataset on assessing work-related mental health among waiter.**
(SAV)

## Author Contributions

**Conceptualization:** Farrukh Ishaque Saah, Hubert Amu, Kwaku Kissah-Korsah.

**Data curation:** Farrukh Ishaque Saah, Kwaku Kissah-Korsah.

**Formal analysis:** Farrukh Ishaque Saah, Hubert Amu, Kwaku Kissah-Korsah.

**Funding acquisition:** Farrukh Ishaque Saah, Hubert Amu, Kwaku Kissah-Korsah.

**Investigation:** Farrukh Ishaque Saah, Hubert Amu, Kwaku Kissah-Korsah.

**Methodology:** Farrukh Ishaque Saah, Hubert Amu, Kwaku Kissah-Korsah.

**Project administration:** Farrukh Ishaque Saah, Hubert Amu, Kwaku Kissah-Korsah.

**Resources:** Farrukh Ishaque Saah, Kwaku Kissah-Korsah.

**Software:** Farrukh Ishaque Saah, Kwaku Kissah-Korsah.

**Supervision:** Hubert Amu, Kwaku Kissah-Korsah.

**Validation:** Farrukh Ishaque Saah, Kwaku Kissah-Korsah.

**Visualization:** Farrukh Ishaque Saah, Kwaku Kissah-Korsah.

**Writing – original draft:** Farrukh Ishaque Saah, Hubert Amu.

**Writing – review & editing:** Farrukh Ishaque Saah, Hubert Amu, Kwaku Kissah-Korsah.

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
