## [Decision Letter · Decision Letter 0]

17 Feb 2021

PONE-D-20-38184

Prevalence and predictors of work-related depression, anxiety, and stress among waiters: A cross-sectional study in upscale restaurants

PLOS ONE

Dear Dr. Saah,

Thank you for submitting your manuscript to PLOS ONE. After careful consideration, we feel that it has merit but does not fully meet PLOS ONE’s publication criteria as it currently stands. Therefore, we invite you to submit a revised version of the manuscript that addresses the points raised during the review process.

Please, be aware that submitting a revision does not guarantee acceptance.

We look forward to receiving your revised manuscript.

Kind regards,

Vincenzo De Luca

Academic Editor

PLOS ONE

Journal Requirements:

Reviewers' comments:

Reviewer's Responses to Questions

**Comments to the Author**

1. Is the manuscript technically sound, and do the data support the conclusions?

Reviewer #1: Partly

2. Has the statistical analysis been performed appropriately and rigorously? 

Reviewer #1: Yes

3. Have the authors made all data underlying the findings in their manuscript fully available?

Reviewer #1: Yes

4. Is the manuscript presented in an intelligible fashion and written in standard English?

Reviewer #1: Yes

5. Review Comments to the Author

Reviewer #1: The present cross-sectional study recruited 384 waiters in upscale restaurants in the Accra Metropolis in Ghana. However, there were several limitations as follows:

1. The sample is not representative enough neither for all restaurants waiters in Accra, nor for upscale restaurants waiters in other areas in Ghana, although authors stated that 6 restaurants were selected by a lottery method from 18 upscale restaurants in the Accra. Besides, authors stated that waiters who met for the inclusion criteria were randomly approached during the day’s work, how many of them agreed with and completed the survey, and how many of them rejected or did not complete the survey? It is inappropriate to make a conclusion of 'high prevalence of poor mental health among waiters' based on the findings obtained from an unrepresentative sample. Furthermore, it is farfetched to associated the limited findings with the SDG.

2. Authors listed a formula for calculating sample size, however, authors did not provide any statistics or parameters of interested variables. Thus, how could author estimate an reasonable sample size?

3. How to identify the depression, anxiety, and stress are work-related or non-work-related?

4. What are the psychometric performance of the scales in the sample included in the present study?

6. PLOS authors have the option to publish the peer review history of their article (what does this mean?). If published, this will include your full peer review and any attached files.

Reviewer #1: No

---

## [Author Response · Author response to Decision Letter 0]

25 Feb 2021

Reviewer #1: The present cross-sectional study recruited 384 waiters in upscale restaurants in the Accra Metropolis in Ghana. However, there were several limitations as follows:

Comment 1: The sample is not representative enough neither for all restaurants waiters in Accra, nor for upscale restaurants waiters in other areas in Ghana, although authors stated that 6 restaurants were selected by a lottery method from 18 upscale restaurants in the Accra. Besides, authors stated that waiters who met for the inclusion criteria were randomly approached during the day’s work, how many of them agreed with and completed the survey, and how many of them rejected or did not complete the survey? It is inappropriate to make a conclusion of 'high prevalence of poor mental health among waiters' based on the findings obtained from an unrepresentative sample. Furthermore, it is farfetched to associated the limited findings with the SDG.

Response 1: Representativeness of the sample has been clarified on page 7 of the manuscript.

Comment 2: Authors listed a formula for calculating sample size, however, authors did not provide any statistics or parameters of interested variables. Thus, how could author estimate an reasonable sample size?

Response 2: The parameters of the interest variables in the sample size formula has been provided on page 7 of the manuscript.

Comment 3: How to identify the depression, anxiety, and stress are work-related or non-work-related?

Response 3: The main assumption of the study is that waiting work in upscale restaurants increases the risk for mental health issues and thus, the study participants were prompted to answer the questions in relation to their work as waiters. This allowed for any observations made to be linked to their job (see page 8).

4. What are the psychometric performance of the scales in the sample included in the present study?

Response 4: The psychometric performance of the scales in the sample in the present study has been included on page 8 of the manuscript

---

## [Decision Letter · Decision Letter 1]

22 Mar 2021

Prevalence and predictors of work-related depression, anxiety, and stress among waiters: A cross-sectional study in upscale restaurants

PONE-D-20-38184R1

Dear Dr. Saah,

We’re pleased to inform you that your manuscript has been judged scientifically suitable for publication and will be formally accepted for publication once it meets all outstanding technical requirements.

Kind regards,

Vincenzo De Luca

Academic Editor

PLOS ONE

Additional Editor Comments (optional):

Reviewers' comments:

Reviewer's Responses to Questions

**Comments to the Author**

1. If the authors have adequately addressed your comments raised in a previous round of review and you feel that this manuscript is now acceptable for publication, you may indicate that here to bypass the “Comments to the Author” section, enter your conflict of interest statement in the “Confidential to Editor” section, and submit your "Accept" recommendation.

Reviewer #1: (No Response)

2. Is the manuscript technically sound, and do the data support the conclusions?

Reviewer #1: Yes

3. Has the statistical analysis been performed appropriately and rigorously? 

Reviewer #1: Yes

4. Have the authors made all data underlying the findings in their manuscript fully available?

Reviewer #1: Yes

5. Is the manuscript presented in an intelligible fashion and written in standard English?

Reviewer #1: Yes

6. Review Comments to the Author

Reviewer #1: Almost all my major concerns have been addressed, except for some minor concerns.

1. It is inappropriate to state that caffeine or alcohol consumption, non-prescription drug or illicit misuse predict or impact poor mental health among the participants, based on findings of a cross-sectional study. As authors recognized in the first sentence in the page 27, waiters with depression, anxiety, or stress might be more likely taking more alcohol, caffeine and non-prescription drugs to alleviate emotional problems.

2. According to the results listed in the tables 4, 5, and 6, associations between religion blief and ethnicity of the waiters and depression, anxiety, or distress did not reach statistical significance. It should be cautious to discuss the associations in the 2nd and 3rd paragraphs in the page 28.

7. PLOS authors have the option to publish the peer review history of their article (what does this mean?). If published, this will include your full peer review and any attached files.

Reviewer #1: No

---

## [Editor Report · Acceptance letter]

6 Apr 2021

PONE-D-20-38184R1 

Prevalence and predictors of work-related depression, anxiety, and stress among waiters: A cross-sectional study in upscale restaurants 

Dear Dr. Saah:

I'm pleased to inform you that your manuscript has been deemed suitable for publication in PLOS ONE. Congratulations! Your manuscript is now with our production department. 

Kind regards, 

on behalf of

Dr. Vincenzo De Luca 

Academic Editor

PLOS ONE